# Delayed Recovery After Exercise-Induced Pain in People with Chronic Widespread Muscle Pain Related to Cortical Connectivity

**DOI:** 10.3390/brainsci14111102

**Published:** 2024-10-30

**Authors:** Mark D. Bishop, Meryl J. Alappattu, Priyanka Rana, Roland Staud, Jeff Boissoneault, Shelby Blaes, Yonah Joffe, Michael E. Robinson

**Affiliations:** 1Department of Physical Therapy, University of Florida, Gainesville, FL 32610, USA; meryl@ufl.edu; 2Center for Pain Research and Behavioral Health, University of Florida, Gainesville, FL 32610, USA; roland.staud@medicine.ufl.edu (R.S.); sblaes@ufl.edu (S.B.); yonahjoffe@ufl.edu (Y.J.); merobin@ufl.edu (M.E.R.); 3Department of Physical Therapy, University of El Paso, El Paso, TX 79968, USA; prana@utep.edu; 4Department of Rheumatology, University of Florida, Gainesville, FL 32610, USA; 5Minnesota Alcohol and Pain Lab, Department of Anesthesiology, University of Minnesota, Minneapolis, MN 55455, USA; jboisson@umn.edu; 6Department of Clinical and Health Psychology, University of Florida, Gainesville, FL 32610, USA

**Keywords:** fMRI, delayed onset muscle soreness, imaging, brain, exercise

## Abstract

Background/Objectives: There is a subset of patients with pain who become worse after exercise. To explore this, we examined the responses of people with chronic primary pain to a standardized high intensity exercise protocol used to induce delayed onset muscle soreness (DOMS). Methods: Ten participants with a diagnosis of chronic widespread muscle pain (CWMP) were matched by age and reported gender to ten participants without muscle pain (i.e., no pain (NP)). Participants completed a standardized DOMS protocol. Pain intensity in the arm at rest and with movement was assessed using daily electronic diaries. Peak pain, the timing of peak pain, and the time to recovery were compared between groups. Associations of pain variables with the functional connectivity of the sensorimotor (SMN), cerebellum, frontoparietal control (FPN), and default mode network (DMN) both within network nodes and the rest of the brain was assessed. Results: Significant differences in peak pain, the time to peak pain, and the time to recovery were noted between groups for both pain at rest and pain with movement after controlling for catastrophizing and pain resilience. Connectivity across the SMN, FPN, and DMN was associated with all pain-related variables. Significant group differences were identified between groups. Conclusions: A standardized muscle “injury” protocol resulted in more pain, a longer time to peak pain, and a longer time to resolve pain in the patient group compared to the NP group. These differences were associated with differences in connectivity across brain regions related to sensorimotor integration and appraisal. These findings provide preliminary evidence of the dysregulation of responses to muscle (micro)trauma in people with chronic pain.

## 1. Introduction

Exercise is recommended as the primary non-pharmacological approach for managing pain conditions, including chronic musculoskeletal pain [1,2]. Recommendations for exercise interventions include aerobic activity and strength training. However, there is a subset of patients with chronic musculoskeletal pain who experience worse muscle pain after exercise [2]. People with chronic widespread muscle pain (CWMP, e.g., widespread muscle pain and fibromyalgia) are one such group of patients who have difficulty participating in exercise intervention programs [3]. Resistance and aerobic exercise engages endogenous pain inhibitory mechanisms in most pain-free individuals [2]; however, people with CWMP do not show this response, suggesting impaired endogenous inhibitory mechanisms [3].

Prior research on individuals without CWMP suggests relationships among cortical regions and pain intensity reported in response to delayed onset muscle soreness (DOMS) after resistance exercise. Resistance exercise that includes eccentric (lengthening) muscle actions in muscles unaccustomed to such forces causes damage to their fibers, resulting in muscle pain, swelling, and weakness. This amalgam of symptoms is often given the broad term DOMS [4,5]. These symptoms and signs typically disappear within several days, after which a patient will progressively begin overloading the musculoskeletal and cardiovascular systems by increasing exercise intensity.

Not all people who perform exercise report DOMS. In previous work, people who did develop DOMS-related pain after exercise had a lower grey matter density in the frontal, occipital, and temporal gyri (left medial and inferior frontal, left middle occipital, left middle temporal, and right superior frontal gyri) compared to those who did not develop pain [6]. Connectivity between these areas and cerebellar network seeds was associated with a lower post-DOMS pain intensity [7], supporting cortical influences on reported pain intensity.

In addition, repeated DOMS-related pain exposure in people without muscle pain is associated with decreasing reports of muscle pain intensity at rest and decreased sensitivity to suprathreshold thermal stimuli after subsequent bouts [8]. These changes are associated with greater adaptation in functional connectivity between the nucleus accumbens–medial prefrontal cortex (NAc-mPFC) and in the sensorimotor network (SMN) connectivity with the dorsomedial, ventromedial, and rostromedial prefrontal cortices. Furthermore, increases in regional signal variability (left lingual gyrus, right MTG, left MTG, and left precuneus) are associated with reductions in pain intensity over time [9]. Combined, these findings suggest that adaptations occur in response to repeated clinically relevant pain among cortical regions involved in endogenous pain modulation, a process considered integral in the effectiveness of exercise as an intervention for pain conditions [2].

Therefore, our purpose was to use DOMS as a clinically relevant probe to investigate endogenous pain modulatory mechanisms in people with CWMP. We hypothesized that people with CWMP would have elevated responses to exercise, delayed recovery after exercise, and differences in functional connectivity (FC) across cortical regions associated with pain intensity compared with pain-free individuals. In addition, we explored how FC among cortical regions was associated with the time to peak pain and time to recovery.

## 2. Materials and Methods

This analysis included people enrolled in an ongoing clinical trial (NCT04441619).

The study recruited individuals (aged 18–70 years) from 2 populations: 10 people with CWMP, e.g., widespread muscle pain or a diagnosis of fibromyalgia (FM) and 10 age- and gender-matched asymptomatic controls. Recruitment occurred through community advertising, rheumatology clinics, and the institutional Consent2Share database of patients who are willing to be contacted about potential research participation. Additional recruiting advertisements were posted to social media sites approved by the Institutional Review Board (IRB) and in collaboration with the Clinical and Translation Science Institute Recruiting Center. Potential participants were directed to a secure REDCap survey. Study staff were alerted by automatic notification of potential interest in the study and then contacted the potential participant through the preferred method indicated in the survey. 

Inclusion criteria for all participants in the parent trial included (a) being between the ages of 18 and 70; (b) not participating in a biceps-specific conditioning program in the past 3 months (the study protocol used high-intensity exercise to induce DOMS as a clinically relevant model of pain; thus participants were excluded if they consistently performed strength training of the target muscle group, i.e., the biceps); and (c) not reporting wrist/hand, elbow, or shoulder pain in the last 3 months.

Potential participants in the group with pain were recruited if they reported WMP and difficulty sleeping or had been diagnosed with FM. Exclusion criteria included chronic medical conditions that may affect pain perception (e.g., numbness in the hands and feet, diabetes, and high blood pressure), kidney dysfunction, muscle damage, a major psychiatric disorder, or any other condition that prevented use of arms or hands; using any interventions (including, but not limited to, massage, medication, and stretching) for symptoms induced by pain training for the duration of the study; having ferromagnetic metal in the head, neck, or abdominal cavity; for female participants, having a positive pregnancy test result. Enrolled participants went through a second screening process for eligibility for magnetic resonance imaging.

In the primary trial, the two participant groups (CWMP) and asymptomatic controls (NP) were randomized to one of three intervention groups: repeated DOMS (RD), single DOMS (SD), and natural history (NH). A second layer of randomization determined which arm a participant exercised first. Randomization was performed by computer software (Research Randomizer www.randomizer.org). This current analysis includes the responses of participants who completed the first session in the RD and SD groups.

### 2.1. Procedures

After completing the informed consent process, participants completed the following measures.

Questionnaires and a resting state functional MRI (rs-fMRI) scan were completed before completing a high-intensity fatiguing isokinetic exercise protocol to induce arm pain. Participants completed online daily pain diaries for fourteen days related to pain generated in the arm following their initial visit.

Questionnaires: Participants completed a demographic intake information form including gender, age, employment status, marital status, educational level, and medical history. In addition, the participants completed the Pain Catastrophizing Scale (PCS) [10], Brief Resilience Scale (BRS) [11], and Pain Resilience Scale (PRS) [12].Structural and functional MRI acquisition: Anatomical and functional (fMRI) data were collected from each participant prior to DOMS induction using a research-dedicated Philips Elition whole-body 3 T scanner. Sequentially, participants had a survey/reference and high-resolution 3D anatomical MRI scan performed first, followed by a resting-state fMRI scan.Experimentally induced pain protocol: Participants performed up to 6 sets of 15 repetitions at a speed of 60°·s^−1^.Daily pain intensity measures: Participants completed daily pain diaries for 14 days using the Research Electronic Data Capture (REDCap) to evaluate the pain in the arm at rest and with movement associated with DOMS following the exercise protocol.

Full descriptions of the measures are included in the Appendix A

### 2.2. Analysis

Summary statistics were calculated for each group (Table 1). Variables that were different between groups at baseline were considered covariates.

Data were downloaded from REDCap and cleaned in Excel before analysis in IBM Statistics (version 29). The peak pain intensity at rest and with movement, time to peak pain intensity (days), and time to recovery (defined as a VAS of <5 out of 100; days) were extracted. The peak pain at rest and with movement, and the time to these peaks were compared between groups using t-tests. The times to recovery were tested using Cox regression and Kaplan–Meier estimators.

Functional MRI preprocessing and the functional connectivity analysis were conducted using the CONN toolbox [13] and SPM12 [14]. Preprocessing steps included realignment, motion and artifact detection, coregistration with structural images, normalization to the MNI space, and smoothing with an 8 mm FWHM Gaussian kernel. Outlier volumes were defined as those where the mean global signal exceeded 5 standard deviations or framewise displacement that exceeded 0.9 mm from the previous image [15]. Denoising steps included the regression of the first 5 principle components of the signal within the white matter and CSF (reflecting physiological noise associated with respiration and cardiovascular activity, respectively), motion parameters and their first order derivatives, and outlier scans, followed by bandpass filtering between 0.008 and 0.09 Hz [16]. Across participants, effective degrees of freedom after denoising ranged from 36.2 to 104.3 (average = 84.3).

The associations between the peak pain at rest and with movement, time to these peaks, and recovery time with FC of the sensorimotor network (SMN), cerebellum, frontoparietal control network (FPN), and default mode network (DMN) both within network nodes and the rest of the brain were assessed (PFDR < 0.05). Network nodes comprising the SMN, FPN, and DMN were defined based on CONN toolbox defaults [13]. SMN nodes included the bilateral premotor/supplemental motor area and primary motor cortex. FPN nodes included the bilateral dorsolateral prefrontal cortices and bilateral angular gyri. DMN nodes included the medial prefrontal cortex, posterior cingulate cortex, and bilateral middle occipital gyri. To minimize extraneous inferences, only pain-related cerebellar structures were used as seeds for cerebellar analyses, including bilateral Crus I, bilateral hemispheric lobule VI, and vermal lobules IV and V.

Visualization of connectivity was completed using MRIcroGL [17] to generate the brain images. 

## 3. Results

Ten people with CWMP and ten without pain (NP) participated. There were no differences in age or sex at birth between groups. The groups differed significantly in pain-related psychological factors, with people with CWMP having significantly higher pain catastrophizing (*p* = 0.002) and lower pain resilience (*p* = 0.001). Refer to Table 1. These factors were included in subsequent regression analyses.

### 3.1. Pain at Rest

Pain intensity in the arm at rest peaked at an average of 20.6 (100 mm VAS) for the NP group and 45.9 for the CWMP group (*p* = 0.026). This occurred 1 day after exercise for the NP group and 3.6 days for the CWMP group (*p* = 0.002). For pain at rest, the Cox regression model was significant (*p* = 0.013), after controlling for PCS and PRS. The group was a significant predictor within the model (*p* = 0.023; Table 2 and Figure 1). The median time to recovery to 50% for the NP group was 2.4 days (95% CI 1.2–3.6) and 9.6 days (95% CI 6.9–12.3) for the CWMP group (Mantel–Cox log rank *p* = 0.001). 

#### 3.1.1. Association of SMN Functional Connectivity with Pain at Rest

Greater connectivity of the SMN with a cluster in right frontal pole predicted a longer time to peak pain at rest, with the CWMP group having significantly greater resting connectivity than the NP group (t18 = 3.30, *p* = 0.004). 

SMN functional connectivity was significantly associated with the recovery time for several clusters. Greater SMN connectivity with clusters with peak voxels in the bilateral postcentral gyrus was associated with faster recovery, while the opposite pattern was detected for clusters with peak voxels in the bilateral frontal pole, right superior frontal gyrus, and left lateral occipital cortex. Individuals with CWMP had significantly lower connectivity of the medial left postcentral gyrus (t18 = −3.53, *p* = 0.002), lateral left postcentral gyrus (t18 = −2.80, *p* = 0.01), medial right postcentral gyrus (t18 = −2.64, *p* = 0.02), and lateral right postcentral gyrus (t18 = −4.00, *p* = 0.001). However, they had significantly greater connectivity of clusters in left frontal pole (t18 = 4.19, *p* = 0.001), right frontal pole (t18 = 3.81, *p* = 0.001), right superior frontal gyrus (t18 = 3.90, *p* = 0.001), and left lateral occipital cortex (t18 = 3.28, *p* = 0.004).

Within the SMN, greater connectivity of primary motor cortex with both the left (b = −0.05, pFDR = 0.03) and right premotor cortex (b = −0.04, pFDR = 0.04) was associated with a faster time to peak pain. Similarly, greater connectivity of the primary motor cortex with the left (b = −0.04, pFDR = 0.003) and right premotor cortex (b = −0.04, pFDR = 0.009) predicted a faster recovery time for pain at rest. Refer to Figure 2. The CWMP and NP groups did not differ significantly in FC between SMN nodes (*p* > 0.33).

#### 3.1.2. Association of Cerebellar Functional Connectivity with Pain at Rest

No significant clusters were detected where cerebellar connectivity was associated with the time to peak pain. However, greater connectivity of cerebellar structures with a large cluster including the bilateral occipital pole, bilateral lateral occipital cortex, bilateral cuneus, and several additional occipital structures was associated with a faster recovery time. Connectivity between these structures was lower in CWMP individuals than NC individuals (t18 = −3.15, *p* = 0.005).

For intra-cerebellar connectivity, greater connectivity between Vermis 4, 5 and left Crus I (b = −0.04, t18 = −3.13, pFDR = 0.02), as well as right lobule VI (b = −0.03, t18 = −2.52, *p* = 0.04), was associated with a faster time to peak pain at rest. Greater connectivity between Vermis 4, 5 and left Crus I (b = −0.03, t18 = −2.91, pFDR = 0.04) was also associated with a faster recovery time. Individuals with CWMP had significantly lower connectivity between Vermis 4, 5 and left Crus I than NC individuals (b = −0.21, t18 = −2.13, *p* = 0.046).

#### 3.1.3. Association of Frontoparietal Network Connectivity with Pain at Rest

No significant associations of FPN connectivity with the time to maximum pain were noted. However, greater FPN connectivity was associated with a shorter recovery time for a cluster centered in the left lateral occipital cortex and a greater recovery time for a cluster centered in the right postcentral gyrus. The CWMP group had significantly lower FPN connectivity for the cluster in the left lateral occipital cortex (t = −3.58, *p* = 0.002) but greater connectivity for the cluster in the right postcentral gyrus (t18 = 4.46, *p* = 0.0003).

Intra-FPN connectivity was not significantly associated with the time to maximum pain or recovery time for pain at rest. Refer to Figure 3.

Specific data and association details are shown in Appendix A.

### 3.2. Pain with Movement

Pain with movement peaked at a median of 25.4 for the NP group and 49.7 for the P group (*p* = 0.038). The median time to peak occurred 1.6 days after exercise for the NP group and 2.3 days for the P group (*p* = 0.072). The Cox regression model was significant (*p* = 0.021) and showed a significant effect of the group (*p* = 0.03; Table 2 and Figure 1), after controlling for PCS and PRS. Kaplan–Meier estimates of the time to recovery to 50% for the NP group was 3.6 (95% CI 0.8–2.2) days and 8.9 (91% CI 5.8–11.9) for the CWMP group (Mantel–Cox log rank *p* = 0.007).

#### 3.2.1. Association of SMN Functional Connectivity with Pain with Movement

Greater connectivity of the SMN with clusters peaking in the right precentral gyrus, left postcentral gyrus, and precuneus was associated with a faster time to peak pain with movement. For each cluster, the mean connectivity was significantly lower in the CWMP group than in the NP group (t18 = −2.43, *p* = 0.03; t18 = −2.48, *p* = 0.02; and t18 = −2.16, *p* = 0.04, respectively; Appendix A).

Greater SMN connectivity was associated with a faster recovery time for clusters with peak voxels in the bilateral postcentral gyri. In contrast, greater SMN connectivity predicted slower recovery for clusters in the left lateral occipital cortex and right superior frontal gyrus. For the medial left postcentral gyrus (t18 = −2.65, *p* = 0.02), lateral left postcentral gyrus (t18 = −2.66, *p* = 0.02), and right lateral postcentral gyrus (t18 = −3.27, *p* = 0.004), CWMP individuals had lower connectivity than NP individuals. The opposite pattern was observed for the left lateral occipital cortex (t18 = 3.12, *p* = 0.006) and right superior frontal gyrus (t18 = 3.09, *p* = 0.006; Figure 2).

For associations of intra-SMN connectivity with pain with movement, greater FC of the left premotor cortex with the right premotor cortex (b = −0.06, pFDR = 0.006) and primary motor cortex (b = −0.06, pFDR = 0.006), as well as the right premotor cortex with the primary motor cortex (b = −0.07, pFDR = 0.004), was also associated with a shorter delay to peak pain and recovery time (b = −0.03, pFDR = 0.046; b = −0.04, pFDR = 0.006; and b = −0.04, pFDR = 0.02, respectively). The CWMP and NP groups did not differ significantly in FC between SMN nodes (*p* > 0.33).

#### 3.2.2. Association of Cerebellar Functional Connectivity with Pain with Movement

No significant clusters were detected where cerebellar connectivity was associated with the time to peak pain with movement. Greater cerebellar connectivity with a cluster peaking in the right lingual gyrus was associated with a faster recovery time for pain with movement. As before, mean cerebellar connectivity with this cluster was lower for CWMP individuals than NP individuals (t18 = −3.01, *p* = 0.008). No associations of intracerebellar functional connectivity with pain with movement were detected (*p* > 0.05). 

#### 3.2.3. Association of Frontoparietal Network Connectivity with Pain with Movement

FPN connectivity was not significantly associated with the time to maximum pain with movement. For the recovery time from pain with movement, greater connectivity with the left lateral occipital cortex and right angular gyrus was associated with faster recovery, while the opposite effect was noted for connectivity with the right postcentral gyrus. The CWMP group had significantly lower connectivity with the left lateral occipital cortex (t18 = −3.61, *p* = 0.002) and right angular gyrus (t18 = −2.89, *p* = 0.01), but greater connectivity with the right postcentral gyrus (t18 = 4.27, *p* = 0.0004; Figure 3).

Regarding intra-FPN connectivity, greater connectivity from the right dorsolateral prefrontal cortex to left (t18 = −3.72, *p* = 0.005) and right (t18 = −3.03, *p* = 0.01) angular gyrus was associated with a shorter time to maximum pain with movement. Greater connectivity between the right dorsolateral prefrontal cortex and left (t18 = −3.32, *p* = 0.001) and right (t18 = −2.79, *p* = 0.018) angular gyrus also predicted a faster recovery time.

Specific data and association details are shown in Appendix A.

## 4. Discussion

People with CWMP had a higher peak pain intensity, longer time to peak pain, and longer time for pain resolution time after exercise-induced muscle injury than people with no pain. These were found for both pain at rest and pain with movement. Consistent with our understanding of chronic pain conditions, people with CMPM had higher pain catastrophizing and lower pain resilience than people without pain. 

An elevated intensity of pain at rest and with movement, as well as a prolonged period of recovery, must be considered when designing management strategies for people with CWMP. Our results suggest people with CWMP may need longer recovery times between bouts/sessions of exercise therapy when participating in a management program that includes exercise interventions. Several participants in the CWMP group continued to experience pain in the arm at rest and with movement almost two weeks after the exercise bout. The finding of delayed recovery is in line with other work. A systematic review of studies of exercise interventions for one type of CWMP, fibromyalgia, reported adverse effects of increased fatigue, muscle pain, and stiffness [18]. Other work indicates people with fibromyalgia may require up to three days of recovery after submaximal exercise [19], and recovery after fatiguing exercise is delayed in people with chronic fatigue syndrome and comorbid fibromyalgia [20]. Combined, these findings have significant implications for rehabilitation and management of CWMP, as the primary non-pharmacological approach recommended for pain management in people with chronic pain is exercise [1].

Our results suggest that management plans must account for prolonged recovery time after exercise when muscle pain is induced either intentionally or unintentionally. The findings, therefore, can guide not only intervention planning but the education of people with chronic primary pain about expected outcomes for the interventions.

Greater connectivity both within the sensorimotor network and between the sensorimotor network and clusters including postcentral gyrus and lateral occipital cortex being associated with lower pain intensity might initially seem counterintuitive given other work indicating decreased connectivity after interventions that reduce pain [21]. However, increased connectivity may reflect more effective pain modulation; that is, greater connectivity within the sensorimotor network and with other structures could be indicative of efficient integration between sensory input and motor output, allowing for more effective pain modulation through top–down control. The postcentral gyrus is crucial for processing somatosensory information, including pain [22]. Increased connectivity between the SMN and this region may also indicate more precise sensory discrimination, allowing an improved appraisal of the sensory experience. Connectivity in this region was associated with peak pain, the time to peak pain and time to recovery for pain at rest and pain with movement. The connectivity in these regions was statistically lower in people with CWMP, highlighting the importance of this finding and supporting differences in appraisal of the sensory information for those people with CWMP.

The FPN is a key network involved in executive functions, cognitive control, attention, and decision-making [23]. Greater connectivity between the FPN and other brain regions suggests enhanced cognitive modulation of pain perception. The right middle temporal gyrus and right angular gyrus are involved in higher-order cognitive processes such as language, memory, and interpretation of sensory experiences [24,25]. The angular gyrus specifically has been described as a brain region involved in combining and integrating sensory input [25]. Greater connectivity between the FPN and these regions may enhance cognitive reappraisal and emotional regulation. Again, people with CWMP had lower connectivity in these regions, suggesting a diminished or modified appraisal and interpretation of the sensory information experienced after pain induction.

The left and right frontal poles are associated with high-level cognitive functions, including future planning, decision-making, and self-regulation [23,26]. Greater connectivity between the FPN and these frontal regions may reflect more strategic and anticipatory thinking about pain, allowing for better planning and management of pain responses. This might involve anticipating the end of pain, developing coping strategies, or engaging in behaviors that reduce pain. Such cognitive strategies can lead to a reduced maximum pain experience by promoting a sense of control and reducing anxiety related to pain. The right superior frontal gyrus is implicated in cognitive control, self-awareness, executive function, and cognitive flexibility. Greater connectivity with the FPN suggests improved cognitive flexibility in response to pain, allowing for adaptive cognitive strategies like shifting attention, reappraising pain, or engaging in relaxation techniques. Enhanced cognitive flexibility can reduce the emotional burden of pain and promote more effective coping mechanisms, leading to a lower pain intensity. Again, those with CWMP demonstrated lower levels of connectivity across these regions.

The regions mentioned (e.g., angular gyrus, middle temporal gyrus, and occipital cortex) are involved in integrating sensory, cognitive, and emotional information. Greater connectivity between the FPN and these areas suggests improved integration of multiple types of sensory information, which can help modulate the pain experience. For instance, integrating sensory input with cognitive control might allow for more effective downregulation of pain. Greater FPN connectivity with these diverse regions suggests ongoing complex and integrative processes related to pain. 

While traditionally known for its role in motor coordination, the cerebellum is increasingly recognized for its involvement in pain processing and modulation [27]. Notably, connectivity among the cerebellum and other regions was not related to the peak pain intensity but to the time to recovery, with greater connectivity among these nodes related to faster recovery from pain. Enhanced connectivity between cerebellar structures and cortical regions such as the occipital cortex and inferior temporal gyrus might therefore reflect a more effective pain modulatory or inhibitory network. The inferior temporal gyrus is involved in higher-order visual processing, memory, and certain cognitive functions related to the interpretation of sensory experiences. Greater connectivity between cerebellar structures and the inferior temporal gyrus could integrate cognitive and emotional processing with pain perception. This integration might allow for more effective top–down modulation of pain. Connectivity among these nodes was related to the duration of the pain experience. People with CWMP had lower connectivity in these regions than people with NP.

## 5. Conclusions

A standardized muscle “injury” protocol resulted in more pain, an increased time to peak pain, and a longer time to resolve pain in people with CWMP compared to those without pain. Connectivity across the SMN, FPC, and cerebellum was associated with all pain-related variables. While the connectivity of SMN and FPC networks with other brain regions related to the peak pain intensity, the time to that peak, and the time to recovery, the connectivity of the cerebellum was only associated with the time to recovery. Connectivity was statistically lower for those with CWMP compared to those with NP across the majority of areas associated with the pain-related variables, providing support for the key roles of these brain regions in the intensity and duration of the pain experience after exercise. 

While robust, these findings must be interpreted in the context that the participants with pain had a chronic primary pain condition, which already implies differences in nervous system function compared to people without pain. Additionally, these data only reflect the responses of 10 people with CWMP, which may hinder direct application to the broader population of people with CWMP.

Nonetheless, statistically reliable results indicate that these findings provide preliminary evidence of CNS dysregulation of responses to exercise in some people with chronic pain. 

## Figures and Tables

**Figure 1 brainsci-14-01102-f001:**
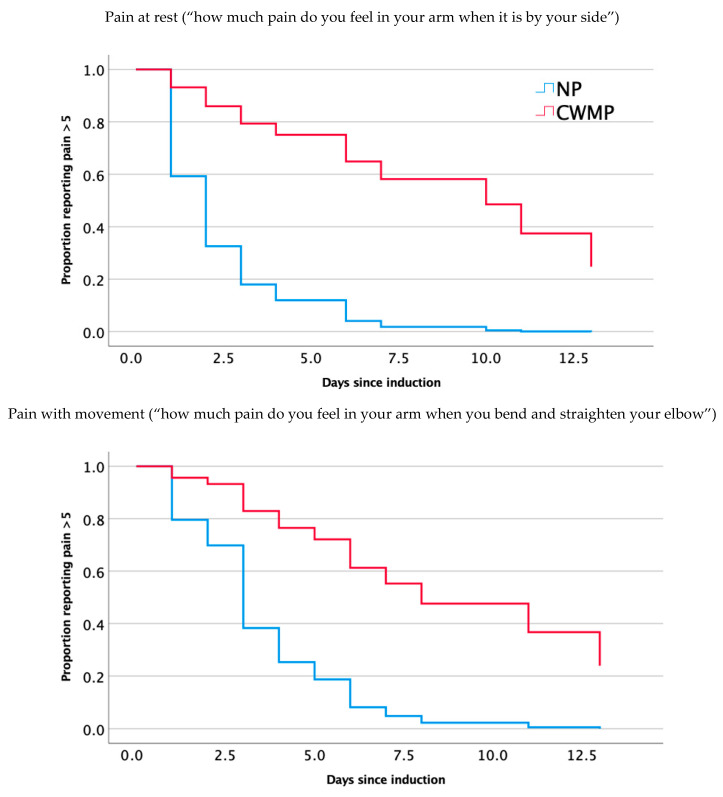
Recovery graphs for pain.

**Figure 2 brainsci-14-01102-f002:**
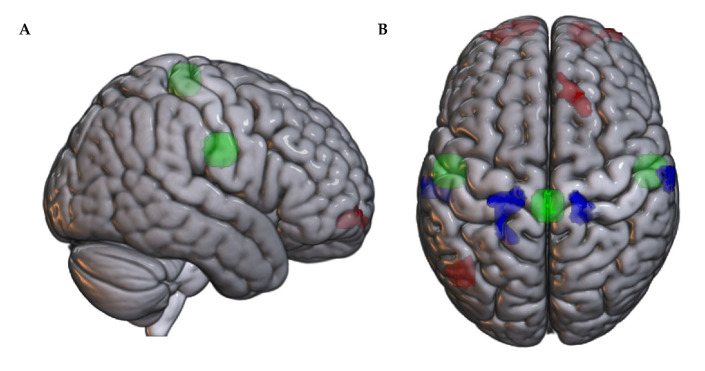
Clusters showing significant associations between connectivity with the SMN (green) and (**A**) the time to maximum pain at rest and (**B**) recovery time. Blue clusters show regions with negative correlation with SMN. Red clusters show regions with positive correlation with SMN.

**Figure 3 brainsci-14-01102-f003:**
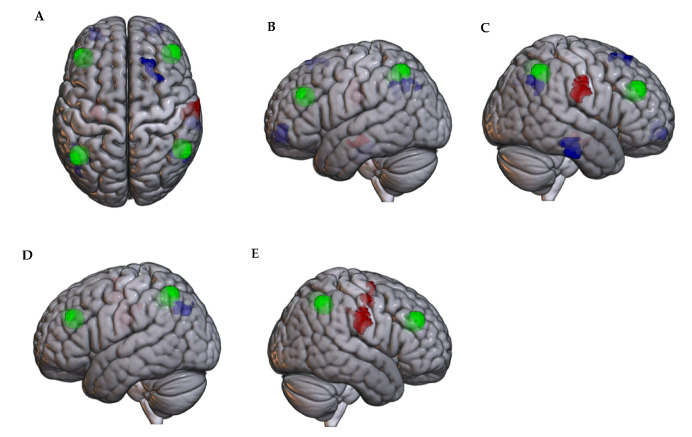
Clusters showing significant associations between connectivity with the FPN (green) and (**A**–**C**) the maximum pain at rest and (**D**,**E**) recovery time. Blue clusters show regions with negative correlation with SMN. Red clusters show regions with positive correlation with SMN.

**Table 1 brainsci-14-01102-t001:** Demographic, psychological, and pain-related variables.

	CWMP		NP		*p*-Value
Age	40.6	16.6	42.3	20.1	0.394
Sex at birth (F, %)	7	70%	6	60%	0.628
Race					
Asian	2		2		
Black	1		1		
White	7		7		
Ethnicity (N identifying as Hispanic)	3		0		0.071
Education					
High School	2		1		
College	5		6		
Graduate	3		3		
	**Mean**	**Std. Deviation**	**Mean**	**Std. Deviation**	***p*-Value**
PCS total	19.5	9.9	7.4	4.4	0.002
Pain Resilience Scale	24.2	7.8	37.6	7.0	0.55
Brief resilience total	3.4	0.9	3.7	0.5	0.26
Max rest	45.9	26.3	20.6	24.7	0.026
Day of peak pain	3.6	2.5	0.8	0.6	0.001
Recovery	8.2	4.8	2.3	2.0	0.002
Max movement	49.7	29.8	25.4	26.3	0.038
Day of peak pain	2.3	1.7	1.6	0.9	0.072
Recovery	7.8	4.7	3.4	2.5	0.009

CWMP—chronic widespread muscle pain; NP—no pain/pain-free; F—female; PCS—Pain Catastrophizing Scale.

**Table 2 brainsci-14-01102-t002:** Cox regression results for the recovery of pain at rest and with movement.

**Pain at Rest**	**−2 Log Likelihood**	**Chi-Square**	**df**	**Sig.**		
First block	65.003	8.407	3	0.038		
Second block		6.139	1	0.013		
Variables	B	SE	Wald	df	Sig.	Exp(B)
PCS	−0.008	0.047	0.028	1	0.868	0.992
PRS	−0.039	0.049	0.633	1	0.426	0.962
Group	2.003	0.883	5.147	1	0.023	7.411
**Pain with Movement**	**−2 Log Likelihood**	**Chi-Square**	**df**	**Sig.**		
First block	66.976	5.986	3	0.112		
Second block		4.493	1	0.034		
Variables	B	SE	Wald	df	Sig.	Exp(B)
PCS	−0.019	0.046	0.166	1	0.683	0.981
PRS	−0.058	0.053	1.192	1	0.275	0.944
Group	1.635	0.83	3.881	1	0.049	5.132

PCS—Pain Catastrophizing Scale; PRS—Pain Resilience Scale.

## Data Availability

The data are not publicly available due to medical privacy laws. Deidentified data are available after reasonable requests by contacting the corresponding author. Requests will be reviewed by the study PI and study team.

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
