# Peer review of "Delayed Recovery After Exercise-Induced Pain in People with Chronic Widespread Muscle Pain Related to Cortical Connectivity"

_brainsci, 2024, doi:10.3390/brainsci14111102_

Round 1

Reviewer 1 Report

Comments and Suggestions for Authors

see attached

Author Response

We have responded to the reviewer's comments within the pdf of the comments.

Reviewer 2 Report

Comments and Suggestions for Authors

The article focuses on the study of the characteristics of delayed recovery after exercise-induced pain in people with chronic widespread muscle pain.

The authors define the goal of the article as follows: to use DOMS as a clinically relevant probe to investigate endogenous pain modulatory mechanisms in people with CWMP.

I believe that the article's goal and title need to be more aligned.

The study helps address a gap regarding how adaptation occurs in response to repeated clinically relevant pain among cortical regions involved in endogenous pain modulation, as well as the gap on how FC among cortical regions was associated with the time to peak pain and time to recovery.

The introduction appears too minimalistic. I recommend that the authors conduct a more in-depth review of the literature.

Table 3 is very large and difficult to interpret. It should either be revised or moved to the appendix.

The software used to create the figures needs to be specified.

The conclusions are consistent with the evidence and arguments presented.

Although the references are appropriate, the authors should expand the range of the literature analyzed.

Author Response

Comment 1: I believe that the article's goal and title need to be more aligned.

We did not modify the title. We did however clarify the purpose statement and change "prolonged responses" to "delayed recovery".

Comment 2: The introduction appears too minimalistic. I recommend that the authors conduct a more in-depth review of the literature.

We have reorganized and added text to the introduction.

Comment 3: Table 3 is very large and difficult to interpret. It should either be revised or moved to the appendix.

We have taken the route of moving the Table to supplemental materials

Comment 4: The software used to create the figures needs to be specified.

The name of the software and the reference for the open source creator has been added to the text.

Comment 5: Although the references are appropriate, the authors should expand the range of the literature analyzed.

We have added additional references to support our premise and the conclusions.

Reviewer 3 Report

Comments and Suggestions for Authors

Thank you for inviting me to review this manuscript.

In this study the authors aim to investigate the responses of people with chronic primary pain to a standardized high intensity exercise protocol used to induce delayed onset muscle soreness (DOMS). The findings indicate initial evidence of dysregulated responses to muscle (micro)trauma in individuals experiencing chronic pain.

The study is potentially interesting, the methods are sounding to me, the work seems well performed and sufficiently detailed, the figures and tables are clear.
Some points remain to be clarified:

·      line 30 of the abstract is missing the point: “..appraisal. These findings…”

·      Age population 18-70: The age range is very wide since this is a study about physical exercise. demonstrate how this aspect does not make the groups too homogeneous and influences the result.

·      The number of subjects included in the groups is missing in the materials and methods paragraph.

·      The sample size is very small.

·      It would be better to add strengths and limitations.

Author Response

Comment 1 - line 30 of the abstract is missing the point: “..appraisal. These findings…”

This has been adjusted.

Comment 2 - Age population 18-70: The age range is very wide since this is a study about physical exercise. demonstrate how this aspect does not make the groups too homogeneous and influences the result.

The study is about the recovery after muscle pain rather than a direct study of exercise. We used a standardized validated procedure to induce the muscle pain and closely match participants by age. By using the standardized protocol and age matching we posit that we are accounting for heterogeniety that might result from any age related differences in the responses to the induction protocol.

Comment 3 - The number of subjects included in the groups is missing in the materials and methods paragraph.

The following text has been added: The study recruited individuals (aged 18-70 years) from 2 populations: 10 people with CWMP, e.g. widespread muscle or a diagnosis of fibromyalgia (FM) and 10 age and gender matched asymptomatic controls.

Comment 4 - The sample size is very small.. It would be better to add strengths and limitations.

While the sample is sample, the matching reduces variability between groups to some degree. in addition there were significant findings with small to moderate effect sizes for both the regression and imaging analyses suggesting that the sample was adequate for our analysis.

The following has been added to the conclusions section: Additionally, these data only reflect the responses of 10 people with CWMP which may hinder direct application to the brader population of people with CWMP.

Round 2

Reviewer 2 Report

Comments and Suggestions for Authors

Dear authors, all of my comments were considered. Thanks.